A revision of Sanpasaurus yaoi Young, 1944 from the Early Jurassic of China, and its relevance to the early evolution of Sauropoda (Dinosauria)

McPhee Blair W. 1 2 blair.mcphee@gmail.com
Upchurch Paul 3
Mannion Philip D. 4
Sullivan Corwin 5
Butler Richard J. 1 6
http://orcid.org/0000-0003-0412-3000 Barrett Paul M. 1 7
1 Evolutionary Studies Institute, University of the Witwatersrand , Johannesburg, Gauteng , South Africa
2 School of Geosciences, University of the Witwatersrand , Johannesburg, Gauteng , South Africa
3 Department of Earth Sciences, University College London , London , United Kingdom
4 Department of Earth Science and Engineering, Imperial College London , London , United Kingdom
5 Key Laboratory of Vertebrate Evolution and Human Origins, Institute of Vertebrate Paleontology and Paleoanthropology, Chinese Academy of Sciences , Beijing , China
6 School of Geography, Earth & Environmental Sciences, University of Birmingham , Birmingham , United Kingdom
7 Department of Earth Sciences, Natural History Museum , London , United Kingdom
Knoll Fabien
Electronic publication date: 2016 Oct 20
Publication date: 2016
Volume: 4
Electronic Location ID: e2578
Received 2016 Aug 2; Accepted 2016 Sep 16
Copyright: © 2016 McPhee et al.
Copyright year: 2016
Copyright holder: McPhee et al.
License: This is an open access article distributed under the terms of the Creative Commons Attribution License, which permits unrestricted use, distribution, reproduction and adaptation in any medium and for any purpose provided that it is properly attributed. For attribution, the original author(s), title, publication source (PeerJ) and either DOI or URL of the article must be cited.
License URL: https://creativecommons.org/licenses/by/4.0/

Keywords: Early Jurassic, China, Middle Jurassic, Sauropoda, Eusauropoda, ‘Vulcanodontidae’

Funding: DST-NRF Centre of Excellence in Palaeosciences, an NRF African Origins Platform 98800 University of the Witwatersrand NRF IRG–China/South Africa Research Cooperation Programme 95449 Marie Curie Career Integration Grant PCIG14-GA-2013-630123 National Geographic Research Wait Grant W421 Royal Society and the Earth Sciences Departmental Investment Fund (NHM) Funding was supplied to B.W.M. by the DST-NRF Centre of Excellence in Palaeosciences, an NRF African Origins Platform (98800) to Jonah Choiniere, and the University of the Witwatersrand. Travel for B.W.M. to China was made possible by the NRF IRG–China/South Africa Research Cooperation Programme (grant no. 95449 to Jonah Choiniere). R.J.B. is supported by a Marie Curie Career Integration Grant (PCIG14-GA-2013-630123). P.U’s research in China was supported by National Geographic Research Wait Grant no. W421. P.M.B’s trips to China have been supported by grants from the Royal Society and the Earth Sciences Departmental Investment Fund (NHM). The funders had no role in study design, data collection and analysis, decision to publish, or preparation of the manuscript.

==============================
The Early Jurassic of China has long been recognized for its diverse array of sauropodomorph dinosaurs. However, the contribution of this record to our understanding of early sauropod evolution is complicated by a dearth of information on important transitional taxa. We present a revision of the poorly known taxon Sanpasaurus yaoi Young, 1944 from the late Early Jurassic Ziliujing Formation of Sichuan Province, southwest China. Initially described as the remains of an ornithopod ornithischian, we demonstrate that the material catalogued as IVPP V156 is unambiguously referable to Sauropoda. Although represented by multiple individuals of equivocal association, Sanpasaurus is nonetheless diagnosable with respect to an autapomorphic feature of the holotypic dorsal vertebral series. Additional material thought to be collected from the type locality is tentatively referred to Sanpasaurus. If correctly attributed, a second autapomorphy is present in a referred humerus. The presence of a dorsoventrally compressed pedal ungual in Sanpasaurus is of particular interest, with taxa possessing this typically ‘vulcanodontid’ character exhibiting a much broader geographic distribution than previously thought. Furthermore, the association of this trait with other features of Sanpasaurus that are broadly characteristic of basal eusauropods underscores the mosaic nature of the early sauropod–eusauropod transition. Our revision of Sanpasaurus has palaeobiogeographic implications for Early Jurassic sauropods, with evidence that the group maintained a cosmopolitan Pangaean distribution.

Introduction

The Early Jurassic was a critical period in the early evolution of sauropod dinosaurs, witnessing the initial radiation of eusauropods and the appearance of several non-eusauropod lineages that did not survive into the Middle Jurassic (e.g., Yates & Kitching, 2003; Upchurch, Barrett & Dodson, 2004; Upchurch, Barrett & Galton, 2007; Allain & Aquesbi, 2008; Yates et al., 2010; Cúneo et al., 2013). However, tracking the early radiation and diversification of Sauropoda has been complicated by its extremely poor early fossil record, with largely incomplete skeletal material from sites that are often imprecisely dated, and compounded by a lack of general consensus regarding the precise diagnosis and definition of Sauropoda (Upchurch, Barrett & Dodson, 2004; Yates, 2007; McPhee et al., 2015a). This is perhaps most evident with respect to the sauropod record from the Early Jurassic of China. Although China is well-known for its diverse array of eusauropod dinosaurs from Middle Jurassic horizons such as the Shaximiao Formation (e.g., Dong, Zhou & Zhang, 1983; Zhang, 1988; He, Li & Cai, 1988; Ouyang, 1989; Pi, Ouyang & Ye, 1996; Peng et al., 2005; Xing et al., 2015), the contribution of the Chinese record to our understanding of basal sauropod evolution remains under-exploited (see Table 1). The stratigraphically lower-most sauropodomorph-bearing horizon within China—the Lower Jurassic Lower Lufeng Formation (Yunnan Province)–while preserving a relative wealth of basal (= non-sauropod) sauropodomorphs, has thus far only produced fossils of equivocal referral to Sauropoda (Dong, 1992; Barrett, 1999; He et al., 1998; Lü et al., 2010) (Fig. 1). For example, the partial skeleton known as ‘Kunmingosaurus’ (Young, 1966; Dong, 1992) still awaits a formal description and diagnosis before its putative basal sauropod status can be confirmed (Upchurch, 1995; Upchurch, 1998; P.M. Barrett, P.D. Mannion & S.C.R. Maidment, 2011, unpublished data). The only other named basal ‘sauropod’ from the Lower Lufeng Formation, Chuxiongosaurus (Lü et al., 2010), appears to be better considered as a non-sauropodan sauropodomorph, similar in general appearance to Yunnanosaurus. The Fengjiahe Formation (Yunnan Province), which is hypothesised to be a lateral equivalent of the Lower Lufeng Formation, has produced the putative basal sauropod Chinshakiangosaurus (Dong, 1992; Upchurch et al., 2007). However, this taxon is known only from a single dentary and partial associated postcranium that, while exhibiting an intriguing mosaic of plesiomorphic and derived features (Upchurch et al., 2007), provides only limited phylogenetic information. Moreover, the whereabouts of the associated post-crania is currently unknown; consequently, character scores for these elements have thus far been based on a small number of published images rather than direct examination of the material (Upchurch et al., 2007). Although better-known than ‘Kunmingosaurus’ and Chinshakiangosaurus, and recovered as a basal sauropod by several recent cladistic analyses (e.g., Yates et al., 2010), the partial skeleton and skull of Gongxianosaurus (Dongyuemiao Member, Ziliujing Formation, Sichuan Province) still awaits a full description (He et al., 1998). In addition, certain aspects of its anatomy (e.g., proportionally low, non-pneumatised dorsal neural arches; three-vertebra sacrum) caution against its inclusion within Sauropoda.

Table 1 Named ‘sauropod’ taxa from the Early Jurassic of China (not including Sanpasaurus).

Taxon	Formation and putative age	Status as sauropod	
Chinshakiangosaurus	Fengjiahe formation	Tentative	
Dong (1992); Upchurch et al. (2007)	?Hettangian		
Chuxiongosaurus	Lower Lufeng formation	Negative	
Lü et al. (2010)	Hettangian–Sinemurian		
‘Damalasaurus’	Duogaila member, Daye Group	Unknown	
Zhao (1985)	?Lower Jurassic		
Gongxianosaurus	Dongyuemiao member, Ziliujing formation	Tentative	
He et al. (1998)	?Toarcian		
‘Kunmingosaurus’	Lower Lufeng formation	Tentative	
Dong (1992)	Hettangian–Sinemurian		
Tonganosaurus	Yimen formation	Positive	
Li et al. (2010)	?Lower–Middle Jurassic		
cf. Eusauropoda	Lower Lufeng formation	Tentative	
Barrett (1999)	Hettangian–Sinemurian		
‘Zizhongosaurus’	Daanzhai member, Ziliujing formation	Positive	
Dong, Zhou & Zhang (1983)	?Toarcian/Aalenian–Bajocian		

Figure 1 Geographic and stratigraphic provenance of Sanpasaurus.

(A) Location of Weiyuan Region within Sichuan Province, People’s Republic of China; (B) Generalized stratigraphic relationships of Early and early Middle Jurassic Chinese sauropodomorphs, based primarily on Dong, Zhou & Zhang (1983), Dong (1992), and Chen et al. (2006). Citations for taxa not mentioned in the text are as follows: Yimenosaurus (Bai, Yang & Wang, 1990), Jingshanosaurus (Zhang & Yang, 1994), and Xixiposaurus (Sekiya, 2010). Geographic details of Sichuan supplied by Map data © 2016 Google.

Several other sauropod taxa named from the ‘Early’ Jurassic of China appear appreciably more derived than those already mentioned and, for this reason, we recommend caution in accepting the current age estimates for these units. This comment is especially salient with respect to Tonganosaurus from the Yimen Formation of Sichuan Province, which has been assigned to Mamenchisauridae (Li et al., 2010), a group otherwise restricted to the Middle–Late Jurassic (Xing et al., 2015). Material assigned to ‘Zizhongosaurus’ (known primarily from a well-laminated partial dorsal neural arch with an anteroposteriorly compressed neural spine) from the Daanzhai Member of the Ziliujing Formation has often been noted as Early Jurassic in age, but potentially dates to the early Middle Jurassic (Dong, Zhou & Zhang, 1983). Relatively little recent study has been carried out on the precise ages of these various Early–Middle Jurassic terrestrial units and more work is needed to establish inter- and intrabasinal correlations between them.

In 1944, C.C. Young described an assemblage of material collected from several quarries in the Maanshan (= Ma’anshan) Member of the Ziliujing Formation close to the town of Changshanling, near Weiyuan City in Sichuan Province. Young (1944) named this material Sanpasaurus yaoi and originally interpreted it as the remains of an ornithopod ornithischian. However, subsequent investigations suggested that at least some of this assemblage was composed of a small-bodied (possibly juvenile) sauropod dinosaur (Rozhdestvensky, 1967; Dong, Zhou & Zhang, 1983; Dong, 1992). Although its sauropod affinities have since been accepted by some authors (but see Weishampel et al., 2004), Sanpasaurus has been largely ignored in the recent literature, and was listed as a nomen dubium by Upchurch, Barrett & Dodson (2004). The Maanshan Member lies directly above the Dongyuemiao Member (from which the remains of Gongxianosaurus were derived and which itself is situated directly above rocks potentially dating to the earliest Jurassic, the Zhenzhuchong Formation) and below the ‘Zizhongosaurus’-bearing Daanzhai Member. Consequently, the Sanpasaurus assemblage has the potential to provide new insights into the sauropod fauna of the Chinese Early Jurassic either prior to, or penecontemporaneous with, the origin of Eusauropoda. Here we provide a detailed description of the identifiable material found within this assemblage, followed by an assessment of its monospecificity and potential taxonomic relationships.

Systematic Palaeontology

DINOSAURIA Owen, 1842

SAURISCHIA Seeley, 1887

SAUROPODOMORPHA Huene, 1932

SAUROPODA Marsh, 1878

Sanpasaurus yaoi Young, 1944

Holotype: IVPP V156A (IVPP V156 partim); Disarticulated middle-posterior dorsal vertebral series, consisting of three complete centra with partial neural arches.

Referred material: IVPP V156B (material removed from holotype, IVPP V156 partim); two centra from the dorsal vertebral series, lacking neural arches; two sacral centra from a small individual; an almost complete anterior-middle caudal vertebra; several distal caudal centra; numerous fragmentary rib shafts; proximal chevron; scapular remains from at least three different elements, potentially including the left and right elements of a single individual; a partial left forelimb consisting of the distal half of a humerus, complete ulna and radius, and the proximal half of a single metacarpal; a femoral head from a small individual; a small ?distal tibia; a proximal fibula; a non-first digit pedal ungual. (N.B. Confusingly, Young noted that the humerus was missing in his original description of Sanpasaurus, but it is figured in Plate I (Young, 1944). As the humerus referred herein matches that figured by Young, we assume that it was relocated subsequent to his publication).

Comments: The majority of the specimens are consistent in preservation—being pale, chalky-brown in color and relatively smooth in texture. This provides some support for Young’s (1944) assertion that at least a subset of the material was discovered in association. However, other included specimens differ from this in being more abraded and somewhat darker in colour. This raises the possibility that IVPP V156 might have been collected from at least two different localities. Moreover, Young (1944) stated that when he received this material some of the labels had been mixed up, as it formed part of a shipment that also contained specimens from other localities around Weiyuan. This suggests caution is warranted with respect to the presumed association of IVPP V156 (Table 2).

Table 2 Select measurements of Sanpasaurus (in mm).

Holotype	
IVPP V156AI	
Anteroposterior length of centrum	103	
Anterior height of centrum	73	
Transverse width anterior centrum face	72	
Neural arch width across parapophyses	96	
IVPP V156AII	
Anteroposterior length of centrum	100	
Anterior height of centrum	82	
Transverse width anterior centrum face	75	
Material potentially associated with holotype on grounds of either size and/or preservation	
Anterior caudal vertebra (IVPP V156B)	
Anteroposterior length of centrum	84	
Anterior height of centrum	103	
Transverse width anterior centrum face	94	
Humerus (IVPP V156B)	
Length as preserved	310	
Minimum shaft circumference	262	
Distal end mediolateral width	155	
Anteroposterior length of ulnar conyle	85	
Ulna (IVPP V156B)	
Maximum length	440	
Maximum transverse width proximal end	135	
Minimum shaft circumference	166	
Anteroposterior length distal end	56	
Transverse width distal end	85	
Radius (IVPP V156B)	
Maximum length	∼425	
Mediolateral width of proximal end	93	
Anteroposterior length of proximal end	53	
Minimum shaft circumference	141	
Mediolateral width of distal end	76	
Anteroposterior length of distal end	57	
Pedal ungual (IVPP V156B)	
Transverse width of proximal end	63	
Dorsoventral height of proximal end	39	
Proximodistal length as preserved	78	
Material of less confident association	
Proximal femur (IVPP V156B)	
Length as preserved	137	
Transverse width across proximal end	175	
Anteroposterior depth of proximal end	86	
Probable distal tibia (IVPP V156B)	
Total length as preserved	138	
Transverse width distal end	130	
Anteroposterior width distal end	68	

In addition, on the basis of size, more than one individual is catalogued within IVPP V156—potentially as many as four on the basis of isolated scapulae (see below). This, and the lack of clear evidence for association between the included elements, renders the taxon unstable, although at least some of the material appears to be taxonomically diagnostic. To protect the taxonomic stability of this species, we hereby restrict the holotype to three dorsal vertebrae, which bear clear autapomorphies that enable it to be diagnosed adequately. Henceforth, we designate the holotype as IVPP V156A. The other material included within IVPP V156 is regarded as potentially referable to the same taxon (see below), but to different individuals and is designated IVPP V156B. This action complies with Article 73.1.5 of the International Code of Zoological Nomenclature (International Commission on Zoological Nomenclature, 1999) in defining the content of the holotype and conferring taxonomic stability.

Diagnosis: Sanpasaurus can be diagnosed by the following autapomorphy: middle-posterior dorsal neural arches with thin, dorsoventrally oriented ridges on the lateral surfaces of the arch, at approximately the anteroposterior mid-point, just above the neurocentral suture. Additionally, following the referral above, Sanpasaurus could be diagnosed by a second potential autapomorphy of the humerus: a distinct midline protuberance between the ulnar and radial condyles.

Locality and horizon: The material was collected from the Maanshan Member of the Ziliujing Formation, Weiyuan region, Sichuan Province, People’s Republic of China in 1939 (Young, 1944; Dong, Zhou & Zhang, 1983) (Fig. 1). Dong, Zhou & Zhang (1983) noted that Dong confirmed this via a prospecting trip in 1978 during which an ungual and vertebral material closely matching that of Sanpasaurus were recovered, though the whereabouts of this additional material is currently unknown. The Ziliujing Formation has been considered to be late Early Jurassic in age (Dong, Zhou & Zhang, 1983; Wang & Sun, 1983; Chen et al., 2006), and the underlying Gongxianosaurus-bearing Dongyuemiao Member has been regarded as Toarcian in age (Meng, Li & Chen, 2003). If the latter is accurate, then the age of the Maanshan Member is no older than the late Early Jurassic.

Previously referred material: In addition to IVPP V156, Young (1944) referred remains (IVPP V221 and V222) from two nearby localities to Sanpasaurus yaoi, and regarded two isolated vertebrae (catalogue numbers unknown) from the Ziliujing Formation near to Chongqing as cf. Sanpasaurus yaoi. Young & Chow (1953) referred another specimen (IVPP V715) from near Chongqing to cf. Sanpasaurus yaoi, although the stratigraphic unit of this locality is unknown. Lastly, Dong (1992: 51) mentioned the discovery of “three incomplete small sauropod skeletons” in the Maanshan Member of Chongqing in 1984 which were suggested to represent Sanpasaurus; however, no further information has been published on this material. Based on a lack of overlapping diagnostic elements, none of these remains can be confidently referred to Sanpasaurus, and we regard them as indeterminate sauropods, restricting Sanpasaurus yaoi to IVPP V156.

Description

Middle-posterior dorsal vertebrae (IVPP V156A)

The newly restricted holotype of Sanpasaurus is composed of three dorsal vertebrae with partially preserved neural arches. The most complete is referred to as V156AI (Fig. 2), whereas the other, less complete vertebrae, are referred to as V156AII (Fig. 3) and V156AIII (Fig. 4), respectively.

Figure 2 Dorsal vertebra (IVPP V156AI).

(A) Anterior view; (B) posterior view; (C) dorsal view; (D) left lateral view; (E) right lateral view. Abbreviations: cdf, centrodiapophyseal fossa; cpol, centropostzygapophyseal lamina; lar, lateral ridge; ms, midline septum; pp, parapophyses; prpl, prezygoparapophyseal lamina; prz, prezygapophyses; tprl, intraprezygapophyseal lamina. Scale bars equal 5 cm. Photographs by B.W.M. and C.S.

Figure 3 Dorsal vertebra (IVPP V156AII).

(A) Anterior view; (B) posterior view; (C) left lateral view; (D) right lateral view. Abbreviations: cpol, centropostzygapophyseal lamina; lar, lateral ridge; tpol, intrapostzygapophyseal lamina. Scale bars equal 5 cm. Photographs by B.W.M. and C.S.

Figure 4 Dorsal vertebra (IVPP V156AIII).

(A) Anterior view; (B) posterior view; (C) left lateral view; (D) right lateral view. Abbreviations: cpol, centropostzygapophyseal lamina; lar, lateral ridge; nc, neural canal; pp, parapophyses. Scale bars equal 5 cm. Photographs by B.W.M.

The centra are mostly intact, whereas the neural spines, postzygapophyses, and diapophyses are missing in all specimens. V156AI preserves both the base and anterior portions of the neural arch, including most of the left prezygapophysis. V156AII is represented primarily by the posteroventral corner of the neural arch, although the ventral part of the anterior surface of the neural arch is also present. V156AIII preserves the right half of the neural arch to the level of the parapophysis. Due to the marked dorsal displacement of the parapophyses (being located well above the neurocentral suture), it is clear that these specimens derive from at least the middle part of the dorsal series.

The centra are amphiplatyan, with a shallowly concave or irregularly flat anterior articular surface and a concave posterior surface. The ventral surfaces are broad and gently convex transversely, rounding smoothly into the lateral surfaces with no distinct ridges. The lateral surfaces have shallow depressions, but no true pleurocoels. This absence is a common feature in the middle-to-posterior dorsal vertebrae of most basal sauropods (e.g., Tazoudasaurus (Allain & Aquesbi, 2008); Shunosaurus (Zhang, 1988); Jobaria (Sereno et al., 1999)). The anteroposterior length of the centrum of V156AI is 1.4 times the height of the anterior surface of the centrum. This is a relatively high ratio, contrasting with 0.96 (middle dorsal) and 0.74 (posterior dorsal) in Tazoudasaurus (Allain & Aquesbi, 2008), and 0.76 (posterior dorsal) in Spinophorosaurus (Remes et al., 2009). By contrast, Shunosaurus appears to have retained relatively elongate centra into the posterior dorsal series, with a length/height ratio of ∼1.2 (Zhang, 1988: Fig. 32). As neither of the isolated dorsal centra (see above) display any marked anteroposterior shortening, it is possible that all elements come from either the anterior or middle part of the dorsal series, or that marked anteroposterior shortening of the dorsal centra did not occur along the dorsal sequence in Sanpasaurus.

The suture dividing the centrum from the neural arch is still clearly visible in all three specimens as a flat, non-interdigitated connection. Although the arch and centrum were clearly semi-fused at the time of death, the apparent lack of complete fusion potentially indicates that the relatively small size of the vertebrae is due to either juvenile or subadult status.

The neural arches appear to have been relatively tall, potentially reaching >1.5 times the height of their respective centra (neural spines excluded). This is a derived sauropodomorph feature and is observed in most basal sauropods (e.g., Tazoudasaurus (Allain & Aquesbi, 2008)). The neural canals are slot-shaped, being considerably taller dorsoventrally than transversely wide. A vertically elongate projection on the anterolateral margin of the neural arch of V156AI is interpreted as the parapophysis and lies at approximately arch midheight or slightly higher. The base of the parapophysis lies just below the level of the dorsal extreme of the neural canal. The arch extends well above the top of the neural canal and it seems that the anterior surface of the arch was shallowly excavated. Two small, parallel ridges extend dorsally across the anterior surface of the arch, beginning at the dorsal opening of the neural canal and possibly extending to the ventromedial corner of each prezygapophysis. These structures, interpreted herein as the intraprezygapophyseal laminae (TPRLs sensu Wilson (1999)) are only minimally separated from one-another with respect to the midline of the anterior surface. Similar, albeit slightly more widely-spaced, TPRLs are potentially present within a posterior dorsal vertebra of Tazoudasaurus (Allain & Aquesbi, 2008: Fig. 14A). The area between the left TPRL ridge and the left parapophysis is moderately excavated, forming a shallow centroprezygapophyseal fossa (CPRF sensu Wilson et al. (2011)). A rounded ridge extends anterodorsally from the top of the parapophysis, forming the anterolateral margin of the arch. This ridge represents the prezygoparapophyseal lamina (PRPL) and is relatively complete apart from the missing anterior tip of the prezygapophysis. A second thinner, sharper ridge extends posterodorsally and would have perhaps joined the dorsal margin of the parapophysis to the ventral margin of the diapophysis as the paradiapophyseal lamina (PPDL). Posterior to this lamina, on the lateral surface of the arch, there is a deep excavation (centrodiapophyseal fossa (CDF)), observable on both sides of V156AI. Internally, the left and right excavations are separated along the sagittal midline of the element by a thin, bony septum. This morphology is potentially homologous to the lateral excavations (= ‘neural cavity’) observed in several other basal sauropod genera (e.g., Barapasaurus, Cetiosaurus, Patagosaurus; see Bonaparte (1986) and Upchurch & Martin (2002: 1059) for discussion). In contrast, although a CDF is commonly observed directly ventral to the diapophysis in most sauropodomorphs (Wilson et al., 2011; Yates, Wedel & Bonnan, 2012), this feature rarely invades the neural arch body to the extreme extent observed in IVPP V156AI.

As mentioned above, the base of the left prezygapophysis is preserved in V156AI, including what appears to be the posterior part of the flattened articular surface and the wall of the hypantrum. If this identification is correct, the prezygapophyseal articulation would have faced inwards at an angle of about 45° to the horizontal. The prezygapophyses appear to have been positioned very close to each other with respect to the midline. The beginning of a ridge extends backwards from the posterodorsal base of the prezygapophysis—towards either the diapophysis or the base of the neural spine (in the case of the former it would be the prezygodiapophyseal lamina (PRDL), in the latter the spinoprezygapophyseal lamina (SPRL)). There is a vertical ridge along the midline of the posterior surface of the neural arch of V156AII, extending dorsally from the roof of the neural canal opening. This potentially represents either the intrapostzygapophyseal lamina (TPOL) or the broken ventral base of the hyposphene (although neither is entirely mutually exclusive). V156AII and V156AIII also preserve the bases of the centropostzygapophyseal laminae (CPOLs). In V156AII these structures bracket either side of the TPOL and are directed steeply posteroventrally, forming the posterolateral margins of the neural arch. The right CPOL of V156AIII is more complete dorsally than in V156AII, and undergoes a marked anteroposterior compression at the level of the dorsal extent of the parapophysis. This narrow lamina forms the posterior wall of a deep, possibly natural, fossa that is walled medially by a thin ridge of bone similar to the median septum observed in V156AI.

In all specimens an unusual structure is present on the lateral surfaces of the neural arch. In V156AI and V156AII it consists of two short and low ridges, subparallel to each other, that extend vertically to produce a low scar or prominence. The dorsal termination of these ridges is roughly level with the ventral termination of the parapophyses, and the ridges themselves are approximately equidistant between the anterior and posterior margins of the neural arch. In V156AIII there is a single ridge that has a more posterodorsal inclination (although only the right lateral surface is preserved), which merges ultimately with the CPOL at roughly the level of the dorsal apex of the neural canal. No similar structures appear to be present in any other Early–Middle Jurassic sauropods, and we provisionally regard the presence of these ridges as an autapomorphy of Sanpasaurus.

Two isolated dorsal centra (IVPP V156B)

In addition to the holotypic dorsal elements (see below) there are two isolated dorsal centra amongst the IVPP V156 assemblage. Both agree in general morphology: the anterior surfaces are nearly flat whereas the posterior surfaces are concave. Both appear to be slightly longer anteroposteriorly than dorsoventrally high or transversely wide (see also below). Their ventral surfaces are concave longitudinally due to the expansion of the anterior and posterior articular surfaces, but are mildly convex transversely. Neither of the dorsal centra possess a sharply-lipped lateral fossa (= pleurocoel). However, one of the centra, possibly from the anterior part of the dorsal series, possesses moderately deep lateral depressions, just posterior to the anterior surface (Fig. 5). On account of these depressions, the lateral and ventral surfaces meet each other abruptly along a rounded ridge that is more developed than that observed in any other dorsal centrum within the assemblage.

Figure 5 ?Mid-anterior dorsal centrum (IVPP V156B).

(A) Left lateral view; (B) ventral view. Scale bar equals 5 cm. Photographs by B.W.M.

Left dorsal rib (IVPP V156B)

A proximally and distally incomplete left thoracic rib is preserved in five pieces (Fig. 6). The tuberculum and capitulum are missing, but the broken proximal portion shows the rib starting to expand into the proximal plate. A groove extends ventrally along the posterior surface throughout most of the proximal half of the preserved length, formed largely by a plate-like ridge that extends along the posterolateral margin and that projects posteriorly. This ridge therefore makes the lateral surface of the rib wider anteroposteriorly. The cross section below the proximal end can thus be described as ‘P’-shaped, with the stem of the ‘P’ formed by the posterolateral ridge or plate, and the rounded part of the ‘P’ formed by the main body of the rib. The anterior surface has a very shallow concavity extending ventrally across its surface, bounded laterally and medially by very subtle ridges along the anteromedial and anterolateral margins. The distal portion has an elliptical cross-section with a flattened lateral surface and a more rounded medial surface. There is no indication of pneumaticity.

Figure 6 Dorsal ribs (IVPP V156B).

Abbreviations: lp, lateral plate. Scale bar equals 5 cm. Photographs by B.W.M.

Sacral vertebrae (IVPP V156B)

Although Young (1944) mentioned that IVPP V156 contained at least five sacral vertebrae, only two unambiguous sacral vertebrae could be located (Fig. 7). Of these, only one preserves the remains of a sacral rib. All of the potential sacral material is notably small, and probably does not pertain to the same individual as either the dorsal vertebral or forelimb (see below) material. The centrum of the most complete sacral element is solid, with no lateral or ventral excavations. The articular surfaces are irregular, but appear to have been predominantly flat. The lateral and ventral surfaces merge smoothly into each other, forming a single rounded convex surface. The rib base is situated on the left side of what we interpret as the ‘anterior’ end of the sacral centrum, and extends posterodorsally from the anteroventral corner at a slightly oblique angle. Little detail can be observed, with the exception that the anterior articular surface appears to be larger than the posterior one, but this might be due to damage and the presence of the rib base.

Figure 7 Sacral vertebrae (IVPP V156B).

(A–C) Isolated sacral vertebra in (A) ?anterior; (B) ?left lateral; and (C) ventral views. (D–F) Possible sacral vertebra in (D) anterior/posterior; (E) lateral; and (F) dorsal views. Abbreviation: sr, sacral rib. Scale bars equal 2 cm. Photographs by B.W.M.

Anterior caudal vertebra (IVPP V156B)

This specimen is missing the dorsal apex of the neural spine, the postzygapophyses, and all but the bases of the transverse processes (= caudal ribs) (Fig. 8). The centrum is solid and amphicoelous, with the anterior surface being somewhat more concave than the posterior one. It is essentially subcircular in cross-section throughout, with the lateral and ventral surfaces of the centrum forming a single rounded convexity. The dorsoventral height of the anterior surface is 1.2 times the anteroposterior length of the centrum. This suggests that the element derives from the posterior end of the anterior caudal series, given that the anterior-most caudal vertebrae of most sauropods tend to possess centra that are considerably shorter anteroposteriorly (e.g., the anterior-most caudal vertebrae of Pulanesaura (McPhee et al., 2015a) and Tazoudasaurus (Allain & Aquesbi, 2008) are roughly twice as high as long). There are no grooves, ridges or hollows on the ventral surface. A single large chevron facet is present on the posterior margin of the ventral surface of the centrum, although the right half of this facet encroaches slightly more anteriorly towards the transverse midline of the centrum than the left half. The chevron facet projects anteroventrally to a level slightly below the ventral margin of the anterior articular face.

Figure 8 Anterior caudal vertebra (IVPP V156B).

(A) Anterior view; (B) posterior view; (C) left lateral view. Abbreviations: hyp, hyposphene; prz, prezygapophysis; sprl, spinoprezygapophyseal lamina; tp, transverse process. Scale bar equals 5 cm. Photographs by B.W.M.

The position of the neural arch on the centrum exhibits a strong anterior bias, although it remains set back from the anterior margin by ∼1.5 cm. The bases of the transverse processes extend for a short distance onto the lateral surface of the centrum and are elliptical in cross-section. The prezygapophyses are narrowly spaced and steeply inclined, with the angle of the articular facets being just under 90° from the horizontal. Finely delimited SPRLs connect the posterior ends of the prezygapophyses with the anterior surface of the neural spine. The SPRLs are still observable at the dorsal termination of the broken neural spine. The fossa located at the base of the spine and bounded by these laminae (spinoprezygapophyseal fossa (SPRF) sensu Wilson et al. (2011)) is relatively shallow. Although the postzygapophyses are missing, a pronounced ridge is preserved ventral to each of their broken bases, which extends to the dorsal margin of the neural canal. This suggests that a hyposphene-like structure was retained until at least the middle of the anterior caudal vertebral series. The neural spine is transversely compressed and directed posterodorsally.

Middle–posterior caudal centra (IVPP V156B)

Several relatively complete middle–posterior caudal vertebrae are present, all lacking their neural arches (Fig. 9). The lateral surfaces of the centra converge ventrally to form a blunt midline ridge, although it is not pinched into a keel. The most complete centrum is amphiplatyan to mildly amphicoelous, and is very gently excavated laterally (Figs. 9A–9C). Its dorsoventral height is 0.75 times its anteroposterior length. There is some indication of a small transverse process, suggesting that this is from the distal part of the middle caudal series. This is consistent with its proportions; in contrast, more derived sauropods lose the transverse ribs earlier in the caudal series—with only the anterior-most 15 caudals bearing ribs (e.g., Haplocanthosaurus (Hatcher, 1903)). The larger of the preserved posterior caudal centra lacks any lateral excavations and has a ventral surface that is smoothly convex (Figs. 9D and 9E). Its dorsoventral height is 0.7 times its anteroposterior length.

Figure 9 Isolated caudal vertebrae (IVPP V156B).

(A–C) ?Middle caudal vertebra in (A) anterior; (B) left lateral; and (C) dorsal views. (D, E) Posterior caudal vertebra in (D) lateral; and (E) anterior/posterior views. Scale bars equal 2 cm. Abbreviation: tp, transverse process. Photographs by B.W.M.

Chevrons (IVPP V156B)

A single proximal chevron (Fig. 10) and part of a more distally located shaft are preserved. The former has a well-developed strut of bone proximally bridging the forked arms of the chevron. This distinguishes the element from the chevrons of Shunosaurus, which are unbridged (Zhang, 1988). The proximal surface appears to have been composed of a single large facet that exhibits a subtle anterior slope. The haemal canal is slot-shaped, being taller dorsoventrally than wide transversely. This differs from the triangular haemal canals of more basal sauropodomorph taxa such as Antetonitrus (McPhee et al., 2014). The walls of the haemal canal open onto the posterior surface of the chevron to form an acute lip of 90° or more. In contrast, the walls of the haemal canal merge more gradually with the anterior surface of the chevron. Moreover, a shallow, fossa-like extension of the haemal canal continues down the anterior surface until at least the level of the missing distal half.

Figure 10 Chevron (IVPP V156B).

(A) Anterior view; (B) posterior view; (C) lateral view. Scale bar equals 5 cm. Photographs by B.W.M.

Scapulae (IVPP V156B)

A maximum of four and minimum of three partial scapulae are present. All are fragmentary, although most of the scapular blade of one can be reconstructed (Fig. 11). The preservation and size of this element and another partial blade within IVPP V156B are similar, and these are potentially referable to the same individual. A third scapular fragment is an anteroposteriorly narrow, dorsoventrally complete section from somewhere along the mid-length of the scapular blade. This fragment has different preservational features (being generally more abraded and slightly darker in colour) to the former two and is potentially associated with a wedge of heavily eroded glenoid region that is also present in IVPP V156B (although this might represent a fourth separate element). The following description focuses on the most completely preserved scapular blade.

Figure 11 Scapular blade (IVPP V156B).

Lateral view. Scale bar equals 5 cm. Photograph by B.W.M.

Overall, the scapular blade shares the general morphology seen in basal sauropod taxa such as Vulcanodon (Cooper, 1984) and Shunosaurus (Zhang, 1988). This is supported by the relatively broad ‘neck’ (the area that would have merged with the proximal plate) and the manner in which this appears to have expanded gradually towards the moderately-broadened distal end. As such, neither the ventral nor dorsal scapular margins appear to have been particularly concave in lateral view. In contrast, the scapular blades of more derived sauropods (e.g., Mamenchisaurus (Ouyang & Ye, 2002); Camarasaurus (Wilson & Sereno, 1998)) are relatively attenuated at their base, with a concomitantly pronounced dorsoventral expansion of the distal blade (see also Mateus, Mannion & Upchurch, 2014: Fig. 7). However, poor preservation and the absence of the proximal plate precludes a more detailed assessment of the proportional relationships of the scapula. The lateral surface of the scapular blade is gently convex dorsoventrally, whereas the medial surface is very gently concave. This differs from the basal sauropodomorph condition whereby the medial surface is either flat or slightly convex (e.g., Antetonitrus, BP/1/4952; McPhee et al., 2014).

Distal half of left humerus (IVPP V156B)

The humerus is broken at roughly mid-shaft, just below the level of the deltopectoral crest; however, when viewed laterally, a slight expansion at its proximal termination probably marks the distal-most extent of the deltopectoral crest. The shaft is subelliptical in cross-section with the long-axis of this section angled at roughly 45° with respect to the transverse axis of the distal end (Fig. 12). The anterolateral corner of the mid-shaft cross-section represents the anterior-most point of the ellipse, and is slightly mediolaterally constricted compared to the rest of the shaft, which is relatively broad transversely. In lateral view the shaft bows slightly posteriorly.

Figure 12 Distal half of left humerus (IVPP V156B).

(A) Anterior view; (B) posterior view; (C) lateral view; (D) medial view; (E) proximal view; (F) distal view. Abbreviations: mt, median tubercle; rac, radial condyle; ulc, ulnar condyle. Scale bars equal 5 cm. Photographs by B.W.M.

The anterior surface of the distal end, although shallowly concave, lacks the pronounced depression (= cuboid fossa) of basal sauropodomorph taxa (Remes, 2008). There is a similarly shallow supracondylar fossa on the posterior surface, located approximately 10 cm from the distal margin. No prominent ridges demarcate the supracondylar fossa. The two distal condyles send out small projections from their anterolateral (ulnar condyle) and anteromedial (radial condyle) margins close to the midline. Within the intercondylar space formed by these projections there is another, smaller anterior projection located at roughly the midline of the distal end. These projections recall the ‘accessory condyles’ previously described as unique to Mamenchisaurus and Spinophorosaurus (Remes et al., 2009), although Upchurch, Mannion & Taylor (2015) have demonstrated that these features are present in many non-titanosaurian sauropods. Nonetheless, the median anterodistal projection (= median tubercle) is a potentially unique feature and is regarded as an autapomorphy of Sanpasaurus herein. Consistent with the derived sauropod condition (Remes, 2008; McPhee et al., 2015a), the distal condyles are not greatly expanded transversely, with the transverse width of the distal end being 1.8 times the anteroposterior depth of the ulnar condyle. The ulnar articulation is the larger of the two condyles and projects anteromedially in distal end view. The distal end is rugose and nearly flat, rounding slightly towards the edges, but does not notably expand onto the anterior or posterior surfaces of the shaft.

Left ulna (IVPP V156B)

Although broken at mid-length and missing a small portion from the proximal end of the anterior (= anterolateral) process, the element is mostly complete (Fig. 13). The ulna is highly elongate, resembling the condition in Vulcanodon and more derived sauropods (Cooper, 1984). Measured from the posterior-most margin of the proximal surface to the estimated tip of the anterior process, the proximal end is approximately 0.3 times the total length of the bone. This contrasts with a ratio of approximately 0.4 or greater for most non-sauropodan sauropodomorphs (e.g., Massospondylus [BP/1/4860]; Antetonitrus [BP/1/4952]). Consistent with the morphology of other sauropods, the proximal end of the ulna is triradiate, with shorter and robust medial and lateral (= posterolateral) processes (these are virtually equal in prominence), and a longer and thinner anterior process. The latter curves strongly laterally towards its termination in proximal view. The resulting concavity for the reception of the proximal radius is thus relatively deep, approaching the condition of Camarasaurus, for example (Wilson & Sereno, 1998). The articular surface, at the point where the three proximal processes meet, is mildly domed and appears to lie a little above the rest of the articular surface. Despite this doming, there is little evidence of a ‘prosauropod’-like olecranon process. The proximal surface is pitted and rugose.

Figure 13 Left ulna (IVPP V156B).

(A) Anterior view; (B) posterior view; (C) proximal view; (D) lateral view; (E) medial view. Abbreviations: ap, anterior process; lp, lateral process; mp, medial process; olp, olecranon process; rl, ligamentous attachment for radius. Scale bars equal 5 cm. Photographs by B.W.M.

In medial view, the shaft bows slightly anteriorly. The proximal part of the shaft is subtriangular in cross-section, with flat surfaces facing anteromedially, anterolaterally and posteriorly. At mid-shaft the ulna becomes more elliptical in cross-section, with the long-axis extending anteroposteriorly. The distal part expands lateromedially but does not expand much anteroposteriorly. The distal articular surface appears to be mildly convex and is highly rugose. There is no evidence of either a ridge or double ridge for ligamentous attachments to the radius on the distolateral corner of the shaft. However, there is a prominent bulge on the lateral surface towards the distal end, but it is not clear how much of this feature is real and how much has been caused by repairs to the shaft. The anterior surface of the distal shaft is planar whereas the other surfaces are gently convex.

Left radius (IVPP V156B)

This is probably the corresponding antebrachial element to the left ulna. Although complete, the shaft is broken into three segments, joined together in a nail and socket arrangement (Fig. 14). The imperfect join at the mid-shaft means that a clean match between these parts is not possible.

Figure 14 Left radius (IVPP V156B).

(A) Anterior view; (B) posterior view; (C) medial view; (D) proximal view; (E) distal view. Abbreviations: mp, medial process. Scale bars equal 5 cm. Photographs by B.W.M.

The proximal end is compressed anteroposteriorly and has an oval outline, with the sharper end of the oval forming the medial process. This process extends proximomedially from the articular surface in a manner similar to that observed in Vulcanodon and other sauropods (see Upchurch, Mannion & Taylor, 2015: Fig. 10). An accompanying (if less laterally-projecting) rise in the lateral corner results in a proximal articular surface that is slightly concave with respect to the transverse plane.

The proximolateral corner of the radius has suffered some slight erosion. The medial margin of the shaft is concave, but it is difficult to say to what degree this morphology is exaggerated due to the abovementioned breakage. In contrast, the radius of Vulcanodon appears to exhibit the opposite condition (see Cooper, 1984: Fig. 6). The distal end of the Sanpasaurus radius has a rugose texture and is relatively flat. If this element is correctly interpreted as a left radius, then the distal surface slopes slightly upwards as it approaches the medial margin. This is the opposite condition to most other sauropods, including Vulcanodon, in which the beveled distal end slopes proximally towards the laterodistal margin (Cooper, 1984; Upchurch, Mannion & Taylor, 2015: Fig. 6) (however, it remains possible that this morphology is either the result of, or has been augmented by, plastic deformation experienced by the shaft). In distal view, the radius has a rounded, subtriangular outline, with a relatively straight posterior margin. This is consistent with the morphology of most sauropods in which the distal end of the radius is circular-to-subrectangular with a flat posterior margin (Wilson & Sereno, 1998; see Upchurch, Mannion & Taylor, 2015: Fig. 9). In contrast, the distal end of the radius in most basal sauropodomorph taxa is an anteroposteriorly elongate ellipse with a relatively acute posterior margin (e.g., Aardonyx: BP/1/5379) (N.B. although Wilson & Sereno (1998) inferred the derived condition for Vulcanodon, examination of Cooper (1984: Fig. 6) suggests that this is potentially an artefact of either erroneous or ambiguous orientation, the distal end of Vulcanodon still being strongly subelliptical-to-rectangular in outline as in more basal taxa).

Proximal end of metacarpal ?IV (IVPP V156B)

Approximately one-third to half of the proximal end of the metacarpal is preserved (Fig. 15). It is triangular in proximal view, with two longer sides of subequal length and one shorter one. The general outline recalls the central (digits II–IV) metacarpus of most basal sauropod taxa (e.g., Allain & Aquesbi, 2008: Fig. 24). In lateral view the proximal surface slopes dorsally towards the most acute corner of this triangle. On the edge of the shaft, directly beneath the least acute corner of the proximal triangle, there is a small, dorsoventrally elliptical tuberosity. This likely represents a site of ligamentous attachment within the metacarpus. The shaft strongly tapers distally, and is roughly square-shaped in cross-section.

Figure 15 Metacarpal (IVPP V156B).

(A) Proximal view; (B–D) indeterminate side views. Abbreviations: lt, ligamentous tuberosity. Scale bars equal 2 cm. Photographs by B.W.M.

Proximal end of ?right femur (IVPP V156B)

The femur is clearly from a smaller individual than the forelimb elements. Moreover, poorer preservation, coupled with a slightly darker colouring, suggests that the femur might come from a different locality than the forelimb elements. Although its incompleteness makes identification of the femur difficult, we interpret it as coming from the right side.

The proximal head projects mainly anteromedially in anterior view, as in other basal sauropods (e.g., Isanosaurus (Buffetaut et al., 2000), Spinophorosaurus (Remes et al., 2009)) (Fig. 16). This contrasts with other taxa that display a more medially oriented femoral head resulting in a sharper angle between the proximomedial apex of the shaft and the distolateral corner of the head (e.g., Antetonitrus (McPhee et al., 2014); Vulcanodon (Cooper, 1984)). There is no distinct neck between the head and greater trochanter region. The middle part of the anterior surface is crushed inwards to form a pronounced hollow. Lateral to this hollow there is a distinct step separating the femoral head from the lateral margin of the proximal end. This step, which forms a small platform just below the level of the medial termination of the femoral head, is more developed anteriorly than posteriorly and is interpreted as the greater trochanter, based on the similar morphology present in taxa like Spinophorosaurus (Remes et al., 2009).

Figure 16 Femoral head (IVPP V156B).

(A) ?anterior view; (B) dorsal view. Abbreviations: gt, greater trochanter. Scale bar equals 5 cm. Photographs by B.W.M.

Distal left tibia (IVPP V156B)

We interpret this element as the distal end of a left tibia from a smaller sized animal than the forelimb elements. The distal end expands prominently transversely from a relatively narrow shaft that is subelliptical in cross-section (Fig. 17). The anterior surface is relatively broad and flat whereas the posterior surface is more convexly rounded—consistent with the morphology of sauropodomorph distal tibiae generally. The distal articular surface is eroded, obscuring the morphology of the ankle-articular joint. However, it appears that the anterior ascending process (= lateral malleolus) was strongly laterally offset from the rest of the shaft.

Figure 17 Distal left ?tibia (IVPP V156B).

(A) Anterior view; (B) posterior view; (C) lateral view. Scale bar equals 5 cm. Photographs by B.W.M.

Proximal left fibula (IVPP V156B)

In lateral view, the proximal head of the fibula is roughly hatchet-shaped, with a pointed posteroproximal corner and more gently rounded anterior margin (Fig. 18). Although the latter surface (= the anteroproximal crest) appears to have been slightly modified by erosion, this morphology is consistent with that seen in most sauropodomorph taxa (e.g., Antetonitrus (McPhee et al., 2014); Camarasaurus (Wilson & Sereno, 1998)). The lateral surface of both the head and the preserved segment of the fibular shaft is highly irregular owing to poor preservation, precluding assessment of any natural ridges and/or excavations that might also be preserved. The incompleteness of the shaft also precludes determination of the extent of the lateral migration of the M. iliofibularis attachment scar (i.e. whether or not this is located anteriorly, as in basal sauropodomorphs). The medial surface of the proximal head is highly rugose and pitted. This texture appears to have covered most of the medial surface of the fibular head, extending from the posteroproximal corner in a diagonal line to a point several centimeters proximal to the base of the anteroproximal crest.

Figure 18 Proximal left fibula (IVPP V156B).

(A) Anterior view; (B) lateral view; (C) medial view. Scale bar equals 5 cm. Photographs by B.W.M.

Pedal ungual from the ?left pes (IVPP V156B)

The ungual is complete apart from the loss of its distal tip. It is dorsoventrally flattened, such that the long-axis of its cross-section is transverse throughout its length (Fig. 19). This establishes the ungual as coming from a digit other than the first, given the characteristic scythe-like morphology of the first pedal ungual in sauropods (Upchurch, Barrett & Dodson, 2004; McPhee et al., 2015a). Within Sauropoda, extreme dorsoventral flattening of the (non-first digit) unguals has only previously been described in the Early Jurassic African taxa Vulcanodon and Tazoudasaurus and represents a potential synapomorphy uniting the two within Vulcanodontidae sensu Allain & Aquesbi (2008; but see Discussion, below). In this regard the digit IV ungual of Vulcanodon (Cooper, 1984: Fig. 35) is a close morphological match for IVPP V156B.

Figure 19 Pedal ungual (IVPP V156B).

(A) Dorsal view; (B) ventral view; (C) ?lateral view; (E) proximal view; (F) distal view. Abbreviations: lg, lateral groove; vf, ventral foramen. Scale bars equal 2 cm. Photographs by B.W.M.

The proximal surface is elliptical in outline and deeply concave, largely due to the prominent overhang (‘lappet’) exhibited by its dorsal margin. The dorsal surface is convex transversely and also slightly convex proximodistally. Near each margin is a prominent groove, each extending virtually the entire length of the claw as preserved. The margin with the slightly shallower groove is interpreted as the lateral because it is slightly concave in dorsal view, whereas the other is regarded as medial because it is slightly convex. This suggests that it is a left claw. It is worth noting, however, that if the unguals figured in Cooper (1984) belong with the left metatarsus of Vulcanodon, then the asymmetrical deflection of the distal end is directed medially in that taxon, suggesting that the ungual described here is potentially from the right side. In contrast, the non-first unguals of Tazoudasaurus are symmetrical in dorsal view. The ventral surface of the IVPP V156B ungual is gently convex transversely and arched upwards in lateral view such that it is mildly concave proximodistally. There are two small foramina located at the proximolateral and proximomedial corners of the ventral surface. A similar foramen is potentially present in the ungual of Vulcanodon (Cooper, 1984: Fig. 35l).

Discussion

The new information presented on Sanpasaurus confirms it as a provisionally valid taxon pending the discovery of further associated and/or referable material. Its validity stems from the two above-mentioned autapomorphic features (see Diagnosis) pertaining to the holotypic dorsal vertebral series and referred distal humerus. These features and other taxonomically significant characters are discussed in more detail below. Given that Sanpasaurus was originally interpreted as an ornithopod ornithischian (Young, 1944), and that this claim is still partially reflected in recent taxonomic lists (e.g., Weishampel et al., 2004: 534), it is worth taking systematic account of the elements within the assemblage that could potentially be interpreted as ornithischian in nature. We also assess the impact of Sanpasaurus on our knowledge of the early sauropod record and its palaeobiogeographical signal.

Is ornithischian material present in IVPP V156?

Although Rozhdestvensky (1967) reinterpreted Sanpasaurus as a small sauropod dinosaur, its identification has remained unresolved in the literature, with some authors regarding at least some of the material as referable to an ornithopod (Weishampel et al., 2004). Rozhdestvensky (1967) correctly pointed out that, in addition to the (now Early) Jurassic age inferred for Sanpasaurus being inconsistent with its identification as an iguanodontid, the lateral excavations of the dorsal neural arches are not seen in any ornithopod dinosaur. Although some iguanodontians possess saurischian-like laminae beneath the diapophyses that frame associated fossae (e.g., Barrett et al., 2011), no known ornithischian possesses dorsal neural arches that are laterally excavated to such a degree that all that separates the paired centrodiapophyseal fossae is a thin, bony septum. Instead, this is a feature more typical of eusauropod dinosaurs such as Cetiosaurus, Patagosaurus and Barapasaurus (e.g., Bonaparte, 1986; Upchurch & Martin, 2002; Upchurch & Martin, 2003).

Although incomplete, it is clear that the dorsal neural arches were originally dorsoventrally tall relative to the height of their respective centra—as observed in Sauropoda (Upchurch, Barrett & Dodson, 2004; McPhee et al., 2014). Within Ornithischia, the only taxa that adopt similarly extreme dorsoventral elongation of the neural arches of the dorsal vertebrae are stegosaurs (Galton & Upchurch, 2004; Maidment et al., 2008) (N.B. This refers to the main body of the arch, excluding the neural spines, which can become very elongate in many other ornithischians, e.g., iguanodontian ornithopods (Horner, Weishampel & Forster, 2004; Norman et al., 2004)). The earliest known unequivocal stegosaur occurrence is Huayangosaurus from the Middle Jurassic Shaximiao Formation of China (Zhou, 1984). Although not as elongate as in Stegosaurus (Maidment, Brassey & Barrett, 2015), Huayangosaurus possesses the heightened neural arch proportions typical of the group (Zhou, 1984). Nonetheless, the dorsal vertebrae of Huayangosaurus differ from those of Sanpasaurus (and other sauropods) with respect to: (1) the anteroposterior restriction of their neural arch bases relative to the lengths of their respective centra (in Huayangosaurus the bases of the neural arches are constricted as they approach the centrum and their anterior and posterior margins are deeply concave in lateral view, whereas in sauropods they are unconstricted, occupy more of the dorsal margin of the centrum and have straighter, subparallel anterior and posterior margins); (2) the lack of anterior centrodiapophyseal/centroparapophyseal laminae; and (3) the apparent absence of any pronounced excavations on the lateral surfaces of the neural arch. Consequently, on the basis of these features, the dorsal vertebrae of IVPP V156 can be considered to be unambiguously referable to Sauropoda.

Rozhdestvensky (1967: 556) also stated that the shape of the caudal vertebrae of Sanpasaurus was inconsistent with the more cross-sectionally “trapeziform” caudal vertebrae of ornithopod dinosaurs. To this we can add that the anterodorsally projecting prezygapophyses of the anterior caudal vertebra of Sanpasaurus contrast with the more anteriorly oriented prezygapophyses in Huayangosaurus (Zhou, 1984).

The forelimb represents the most unambiguously non-ornithischian material within the assemblage, clearly belonging to that of a columnar-limbed, parasagittal quadruped (i.e., even most ornithischian quadrupeds, such as stegosaurs, retain a laterally flexed forelimb posture: Maidment & Barrett, 2012). With respect to the proximal femur (which is dubiously associated with the rest of the assemblage), a proximomedially oriented femoral head is distributed throughout both Sauropoda and Ornithischia (e.g., Weishampel, Dodson & Osmólska, 2004). However, its incompleteness precludes further discussion of its affinities.

Prior to the discovery of Vulcanodon from the late Early Jurassic of southern Africa (Raath, 1972), dorsoventrally compressed pedal ungual phalanges would have been seen as the strongest evidence of an ornithischian within the assemblage. Dorsoventrally low pedal unguals occur early in ornthischian evolution (e.g., Scelidosaurus (Owen, 1863); Scutellosaurus (Colbert, 1981)) and persist throughout the remainder of the group’s history, becoming especially marked in derived members of Thyreophora, Ornithopoda, and Ceratopsia (Weishampel, Dodson & Osmólska, 2004). Although some non-sauropodan sauropodiforms possess pedal unguals that are as wide transversely as dorsoventrally high in proximal aspect (e.g., Blikanasaurus, Antetonitrus; see McPhee et al., 2014), the general condition within Sauropoda is that of a large, mediolaterally compressed, scythe-like ungual on the first digit of the pes, with a similar—if less strongly mediolaterally compressed—morphology observed in the remaining digits (e.g., Apatosaurus: Gilmore, 1936). However, beginning with the revised description of Vulcanodon (Cooper, 1984), and followed more recently by the complete description of Tazoudasaurus (Allain & Aquesbi, 2008), it is now clear that dorsoventrally compressed (non-first digit) pedal unguals were present within at least some basal members of Sauropoda. The question then is whether the morphology observed in IVVP V156 is closer to that of basal sauropods or to Early–Middle Jurassic ornithischian taxa? As stated above, the Sanpasaurus ungual is an extremely close morphological match for that of Vulcanodon (Cooper, 1984). This is evinced by the strongly tapered distal end, deep colateral grooves and the small foramina on the proximoventral surface. Furthermore, the relative transverse width and general absolute proportions of the IVPP V156B ungual are suggestive of a heavy-set, graviportal animal—an ecomorphospace exclusively occupied by Sauropoda during the Early Jurassic. Although basal thyreophorans were beginning to enter this ecomorphospace, the pedal unguals of taxa such as Scelidosaurus (Owen, 1863; NHMUK PV R1111) are relatively narrow compared to IVPP V156B (the pedal unguals of the earliest stegosaurs are incompletely unknown: Zhou, 1984). Moreover, the pedal unguals of basal ornithopod dinosaurs are relatively narrow in dorsal view, even if the ventral surface is somewhat broadened (Norman et al., 2004). Given the association of the ungual with a suite of material that is clearly referable to Sauropoda, and its similarity to those of Vulcanodon and Tazoudasaurus, we argue that it is best considered as pertaining to a sauropod.

Phylogenetic affinities

Assessing the phylogenetic position of Sanpasaurus is difficult due to its incompleteness and the ambiguous associations of the type assemblage. It can be referred to Sauropoda based on a number of features that are derived within Sauropodomorpha (e.g., slender ulna with a deep radial fossa (Bonnan & Yates, 2007); advanced laminar configuration of the dorsal vertebrae (e.g., Wilson & Sereno, 1998; Upchurch, Barrett & Dodson, 2004); see Description above). However, determining its affinities within this clade is much more problematic, with several features arguing against its inclusion within Eusauropoda (e.g., all dorsal centra are non-opisthocoelous and lack lateral depressions; dorsoventrally flattened pedal unguals: Wilson & Sereno, 1998; Upchurch, Barrett & Dodson, 2004). The elements that are of greatest diagnostic utility are the dorsal vertebrae with partial neural arches and the pedal ungual. Although the incompleteness of the dorsal vertebrae limits their information content, several features warrant discussion.

The laminae most clearly developed in Sanpasaurus that are absent in non-sauropodan sauropodomorphs (‘prosauropods’) are the TPRLs, PRPLs and TPOLs. Unfortunately, the absence of cervical and dorsal vertebrae in the available material of Vulcanodon limits our understanding of the timing of acquisition of these features. Allain & Aquesbi (2008: Table 2) summarized the distribution of the major laminar structures across several basal sauropod taxa (as well as the neosauropods Apatosaurus and Camarasaurus). Confusingly, the presence/absence of TPRLs in the middle-to-posterior dorsal vertebrae of all included taxa is listed as an inapplicable character (Allain & Aquesbi, 2008: Table 2), probably reflecting Wilson’s (1999: 647) assertion that the TPRL disappears from the ∼fourth dorsal vertebrae onwards as the anterior surface of the neural arch is modified by the hyposphene-accommodating hypantrum. However, Sanpasaurus clearly possesses a set of paired, well-defined ridges extending from the median convergence of the prezygapophyses to the dorsal margin of the neural canal—structures interpreted herein as homologous with the TPRLs sensu Wilson (1999) (in contrast, the CPRLs are more laterally positioned, extending all of the way to the neurocentral junction). Furthermore, examination of a posterior dorsal vertebra of Tazoudasaurus (To1-156, Allain & Aquesbi (2008: Fig. 14); also a high quality colour photograph of the element supplied to B.W.M. by Allain in 2013) suggests that the medial margins of both prezygapophyses were ornamented with finely delineated TPRLs that extend to the dorsal margin of the neural canal in a fashion similar to that in Sanpasaurus. If this interpretation is correct, then TPRLs developed relatively early in sauropod evolution, either becoming lost in the middle-to-posterior dorsal vertebrae of more derived eusauropod taxa, or being modified to a thick, horizontal ridge connecting the prezygapophyses at the rear of the hypantrum (and thus separating the SPRF from the CPRF) (e.g., Haplocanthosaurus (Hatcher, 1903: Plate I); Camarasaurus (Osborn & Mook, 1921)).

Both PRPLs and TPOLs also appear to have developed relatively early in sauropod evolution, being present in all sauropods from Tazoudasaurus onwards (Allain & Aquesbi, 2008), and hence the presence of these features in Sanpasaurus is not particularly informative with respect to phylogenetic relationships (although the condition in Kotasaurus remains ambiguous (Yadagiri, 2001)). With respect to the TPOL, it is worth noting that Wilson (1999: 647) stated that this lamina is also lost in most sauropod taxa with the appearance of the hyposphene at the end of the anterior dorsal series (an exception being diplodocids). However, in taxa that develop relatively attenuated hyposphenes (e.g., Tazoudasaurus (Allain & Aquesbi, 2008); Mamenchisaurus (Ouyang & Ye, 2002); Bellusaurus (Mo, 2013)), this feature can persist well into the posterior end of the dorsal vertebral series. The absence and/or poor development of other common laminae (i.e., CPRL, CPOL, PCPL) in Sanpasaurus might reflect the posterior positioning of the preserved dorsal neural arches within the dorsal series, with some taxa (e.g., Mamenchisaurus: Ouyang & Ye, 2002) exhibiting relatively undeveloped CPOLs in more posterior dorsal vertebrae, whereas in Bellusaurus (Mo, 2013) these structures persist throughout the dorsal series. Likewise, CPRLs in both Tazoudasaurus and Bellusaurus appear more developed in anterior dorsal vertebrae than in posterior ones. Nonetheless, the paucity of well-preserved dorsal vertebral series for the majority of Early Jurassic sauropod taxa precludes further assessment of laminar morphological evolution within the group. This same concern applies to the lack of well-figured information for important Middle Jurassic taxa such as Shunosaurus.

A final point worth mentioning with respect to the dorsal vertebrae of Sanpasaurus is the lateral excavation of the base of the neural arches, which is positioned directly ventral to where the diapophyses would have been located. Although there is no evidence of a Barapasaurus- or Patagosaurus-like cavity within the arch itself, which is linked to the external surface via a lateral foramen (Jain et al., 1979; Bonaparte, 1986), Upchurch & Martin (2003: 218) noted that in some of the dorsal vertebrae in these specimens, and also in Cetiosaurus (Upchurch & Martin, 2002; Upchurch & Martin, 2003), there is a deep pit on either side of the arch which is separated from its partner on the opposite side by a thin midline septum. The presence of a similar feature in Sanpasaurus might indicate that these taxa are related. However, Barapasaurus, Cetiosaurus and Patagosaurus all possess dorsal vertebrae in which at least the anterior-most centra are opisthocoelous. Although no anterior-most dorsal centra are present in IVPP V156 (based on the absence of parapophyses from the centra), all of the centra of Sanpasaurus are amphicoelous, and it is possible that the centrum with a shallow lateral fossa might represent an anterior dorsal vertebra. Furthermore, the lateral centrum surfaces in Barapasaurus, Cetiosaurus and Patagosaurus possess pronounced fossae, if not ‘true’ pluerocoels (i.e., sharp-rimmed, invasive foramina)—a feature not seen in any of the dorsal centra within IVPP V156.

The forelimb is relatively typical for sauropods (Upchurch, Barrett & Dodson, 2004; Remes, 2008). However, at least two features distinguish it from the Early Jurassic taxa Vulcanodon and Tazoudasaurus. As mentioned above, the anterodistal margin of the humerus is ornamented with accessory projections of the distal condyles, a feature common (if variable in expression) to a number of sauropod genera (Remes et al., 2009; Upchurch, Mannion & Taylor, 2015). These features are clearly absent in Tazoudasaurus (Allain & Aquesbi, 2008), and possibly Barapasaurus too (Bandyopadhyay et al., 2010; Fig. 9). Unfortunately, the distal humerus of Vulcanodon is incomplete, precluding comparison with Sanpasaurus. The proximal ulna of Vulcanodon, however, differs in appearance from that of Sanpasaurus in being somewhat transitional between the ‘prosauropod’ condition and that of later sauropods. This is seen in the minimally-deflected, elongate anterior process and comparatively undeveloped lateral process (Cooper, 1984: Fig. 8). In contrast, the proximal ulna of Sanpasaurus exhibits the more typically sauropodan triradiate condition with a laterally curved anterior process. The proximal ulna of Tazoudasaurus is too incomplete to permit comparison (Allain & Aquesbi, 2008: Fig. 22). With these differences in mind, the forelimb morphology of Sanpasaurus appears to have been relatively derived compared to that of Vulcanodon and Tazoudasaurus.

Although differing in forelimb morphology, the most striking similarity between Sanpasaurus, Vulcanodon and Tazoudasaurus is the dorsoventrally compressed non-first pedal ungual. Both Wilson & Sereno (1998) and Upchurch, Barrett & Dodson (2004) suggested that transversely compressed pedal unguals II and III are synapomorphic for Eusauropoda. However, until recently, Vulcanodon possessed the only known unguals for a non-eusauropod sauropod (Cooper, 1984). Confirmation of the same morphology in the (non-first digit) unguals of Tazoudasaurus and Sanpasaurus underscores the extent to which dorsoventral flattening of the unguals appears to have been distributed among basal sauropods (see also Rhoetosaurus (Nair & Salisbury, 2012: Fig. 12) for something of an intermediary morphology). Nonetheless, the absence of this morphology from any taxa more derived than Shunosaurus suggests that transversely compressed pedal unguals can tentatively be considered a genuine synapomorphy of Eusauropoda for the time being (although an ungual collected with material referred to the eusauropod Jobaria also displays this dorsoventrally compressed morphology (MNN TI-22: P.D. Mannion, personal observation, 2013), and thus might indicate a more complicated distribution for this feature).

In summary, it is clear that the IVPP V156 assemblage includes an animal that is transitional between the relatively plesiomorphic morphology of basal sauropods, and the more derived conditions present in eusauropods. The former is supported by the non-opisthocoelous, fully-acamerate condition of the dorsal vertebral centra, the similarities in laminar configuration shared with basal sauropods such as Tazoudasaurus, and the dorsoventrally compressed pedal ungual (see below for discussion regarding the ‘Vulcanodontidae’). An affinity with eusauropods is supported by the (probably) pneumatic excavations of the lateral surfaces of the dorsal vertebral neural arches, and the modifications to the distal condyles of the humerus. Based on these observations, we refer Sanpasaurus to Sauropoda incertae sedis, while highlighting the possibility that Sanpasaurus represents one of the most derived non-eusauropodan sauropods currently known (see also Spinophorosaurus: Remes et al., 2009). Although this possibility could be tested via a cladistic analysis, we have opted to exercise caution in treating IVPP V156 as a distinct operational taxonomic unit due to both its incompleteness and the potentially chimerical nature of the assemblage (thus heightening the possibility of artificially inflating character-conflict within the analysis).

Relevance of Sanpasaurus to basal sauropod palaeobiogeography

The affinities discussed above for Sanpasaurus have implications for the global distribution of basal sauropods in the Early Jurassic. Remes et al. (2009) reviewed the palaeobiogeography of early sauropods and suggested that expansion of the Central Gondwanan Desert during the late Early Jurassic acted as an ecological barrier separating a South Gondwanan clade of Barapasaurus (India) + Patagosaurus (Argentina) from the rest of Eusauropoda. This was not the first time that a form of early sauropod endemism has been hypothesized, with the grouping of Vulcanodon (Zimbabwe) and Tazoudasaurus (Morocco) into the subfamily ‘Vulcanodontidae’ suggestive of an African radiation of basal sauropods (Allain & Aquesbi, 2008). However, both of these interpretations are subject to concerns associated with a poor and patchily sampled fossil record, incomplete taxa, and mutable phylogenetic relationships.

The latter two uncertainties are perhaps best exemplified by the basal position Remes et al. (2009) recovered for Cetiosaurus (United Kingdom) outside of Eusauropoda. This is incompatible with almost all other recent analyses, which place Cetiosaurus well within Eusauropoda, and sometimes as the sister-taxon to Neosauropoda (e.g., Upchurch, Barrett & Dodson, 2004; Upchurch, Mannion & Taylor, 2015; Yates, 2007; McPhee et al., 2014; Otero et al., 2015). Furthermore, placement of Cetiosaurus in a pectinate grade between Vulcanodon and Tazoudasaurus (Remes et al., 2009: Fig. 6) is incompatible with the above-mentioned ‘vulcanodontid’ hypothesis, as well as numerous analyses that find the two Early Jurassic African taxa to be more closely related to each other than either is to the Middle Jurassic Cetiosaurus (e.g., Allain & Aquesbi, 2008; Yates et al., 2010; McPhee et al., 2015a; McPhee et al., 2015b). With respect to the hypothesis of South Gondwanan endemism, it is interesting that Remes et al. (2009: 7) noted that the only unambiguous synapomorphy of a Barapasaurus + Patagosaurus clade is the presence of a “subdiapophyseal pneumatopore,” a feature presumably synonymous with the lateral excavations described above for Sanpasaurus and also present in Cetiosaurus (Remes et al. (2009) also identified the same feature in Tazoudasaurus and Mamenchisaurus; however, although it appears that the former possessed well-developed infradiapophyseal subfossae sensu Yates, Wedel & Bonnan, (2012), the degree to which these structures impacted into the body of the neural arch cannot currently be determined. In contrast, neither lateral excavations nor invasive subfossae of any sort can be confirmed in the one well-figured description of Mamenchisaurus (Ouyang & Ye, 2002); also P. Upchurch & P.M. Barrett, 2010, personal observation). This (now) geographically widespread feature can therefore be regarded as either symplesiomorphic for a wide range of basal sauropods, or highly homoplastic (and likely variable in expression). As a final cautionary note, it is worth mentioning that Barapasaurus is primarily based on a (heavily reconstructed) composite mount from a large bone-bed, the monospecificity of which is yet to be fully demonstrated (see Bandyopadhyay et al., 2010). This, along with the fact that a detailed treatment of the taxonomy and osteology of Patagosaurus is still awaited, clearly limits the utility of these taxa in palaeobiogeographical reconstructions of early sauropod evolution.

Although support for a south Gondwanan basal eusauropod clade is weak, the evidence for a monophyletic radiation at the base of Sauropoda—the ‘Vulcanodontidae’—is somewhat stronger. This is due to a number of similarities between Vulcanodon and Tazoudasaurus (e.g., transverse compression of the tibia; relatively elongate proportions of the pes; dorsoventral flattening of the pedal unguals; Allain & Aquesbi, 2008). Although the sister-taxon relationship between these taxa is sensitive to the position of the highly incomplete Isanosaurus (Buffetaut et al., 2000; see McPhee et al., 2014; McPhee et al., 2015a), and to the inclusion of Spinophorosaurus (Nair & Salisbury, 2012), a close phylogenetic relationship has been resolved in most analyses that have included both African genera (e.g., Allain & Aquesbi, 2008; Otero et al., 2015). The possession of the ‘vulcanodontid’ condition of a dorsoventrally compressed pedal ungual in Sanpasaurus can be interpreted as evidence that either: (1) ‘vulcanodontids’ extended beyond Africa; or (2) that dorsoventrally compressed pedal unguals characterized a wider range of basal sauropod taxa than currently recognized (as is also the case in the lateral excavations on the dorsal neural arches—see above). Given that the limited information currently available for Sanpasaurus suggests a character suite broadly intermediary between basal sauropods and eusauropods, we argue that ‘vulcanodontid’ monophyly in a maximally inclusive sense is probably unlikely—an observation further supported by the depauperate taxonomic content of the proposed subfamily (i.e., two taxa). Nonetheless, additional sampling of the Early Jurassic is required in order to establish a better sense of the phylogenetic distribution of these typically ‘vulcanodontid’ characters.

With respect to the above, and contrary to the scenario posited by Remes et al. (2009), our revision of Sanpasaurus tentatively suggests that early sauropod faunas were probably cosmopolitan throughout Pangaea in the Early Jurassic, with little evidence of geographically-bounded endemism. Although it remains possible that a grade of basal forms originated in Africa prior to its isolation by expansion of the Central Gondwanan Desert, uncertainties remain as to the degree to which aridity could restrict sauropod distributions, with the earliest representatives of the group possibly inhabiting semi-arid environments (e.g., Antetonitrus, Pulanesaura, Vulcanodon: Cooper, 1984; Yates & Kitching, 2003; McPhee et al., 2015a). Nonetheless, the features shared between Sanpasaurus and later near-or-basal eusauropods (e.g., the modifications to the distal humerus) are consistent with Remes et al.’s (2009) observation of a high-degree of faunal exchange between the low-latitude climes of North Gondwana and East and West Laurasia well into the Middle Jurassic. Further exploration and sampling of the Early Jurassic record of China, along with comprehensive reexamination of important Middle Jurassic taxa like Shunosaurus, are necessary to more closely integrate these taxa into overviews of early eusauropod diversification.

Conclusions

Our reassessment of the basal sauropod Sanpasaurus has shown it to be a provisionally valid taxon pending additional sampling of Early–Middle Jurassic strata of China. The unique combination of plesiomorphic and apomophic characters observable in Sanpasaurus underscores the mosaic manner of trait-acquisition that likely characterized the basal sauropod–eusauropod transition. This is perhaps most evident with respect to the presence of dorsoventrally compressed pedal unguals in Sanpasaurus. Whereas the taxa possessing this feature can now be shown to have had a geographic distribution far beyond Africa, its association with eusauropod-like alterations of the dorsal vertebrae and distal humerus also provides additional support to previous assertions of ‘vulcanodontid’ paraphyly (e.g., Upchurch, 1995; Barrett & Upchurch, 2005). Although the incompleteness of this material, coupled with its equivocal association, means that these conclusions must be treated as tentative for the time being, this study also highlights the additional information that can be gleaned from the in-depth re-examination of historically collected and poorly characterized Chinese taxa. Further fossil sampling, as well as the comprehensive reanalysis of other poorly known taxa (e.g., Kunmingosaurus), will be necessary to corroborate the above observations and to better elucidate the contribution of the Chinese Early Jurassic fossil record to our understanding of basal sauropod evolution generally. However, the limited information available from Sanpasaurus provides evidence that at least some sauropod lineages had a global, or near-global, distribution during the Early Jurassic.

We would like to thank Fang Zheng for access to the material and for helpful assistance while undertaking work at the IVPP. We are also grateful to Paul Sereno for allowing us to comment on an unpublished specimen of Jobaria, and to Bob Masek for facilitating access to this material. Greg Edgecombe (NHM) is thanked for valuable taxonomic advice. We also thank Xu Xing, Jingmai O’Connor and CS for their help and hospitality while working at IVPP. Reviews by Toru Sekiya and John Whitlock improved the quality of this contribution.

Institutional Abbreviations

BP Evolutionary Studies Institute (formerly the Bernard Price Institute), University of the Witwatersrand, Johannesburg, RSA

IVPP Institute of Vertebrate Paleontology and Paleoanthropology, Beijing, China

MNN Musée National du Niger, Niamey, Republic of Niger.

Additional Information and Declarations

Competing Interests

Author Contributions

Data Deposition

The authors declare that they have no competing interests.

Blair W. McPhee conceived and designed the experiments, analyzed the data, wrote the paper, prepared figures and/or tables, reviewed drafts of the paper.

Paul Upchurch conceived and designed the experiments, analyzed the data, wrote the paper, reviewed drafts of the paper.

Philip D. Mannion conceived and designed the experiments, analyzed the data, wrote the paper, reviewed drafts of the paper.

Corwin Sullivan contributed reagents/materials/analysis tools, prepared figures and/or tables, reviewed drafts of the paper.

Richard J. Butler analyzed the data, reviewed drafts of the paper.

Paul M. Barrett conceived and designed the experiments, analyzed the data, wrote the paper, reviewed drafts of the paper.

The following information was supplied regarding data availability:

The raw data measurements are shown in Table 2.

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
