# Peer review of "A revision of Sanpasaurus yaoi Young, 1944 from the Early Jurassic of China, and its relevance to the early evolution of Sauropoda (Dinosauria)"

_PeerJ, doi:10.7717/peerj.2578_

## Round 0.1 · original submission · Minor Revisions

I have now received two reports on your manuscript. Both are positive, although they include some suggestions that need to be addressed prior to formal acceptance. Among other things, the reviewers expressed doubts that all the referred material could belong to the same species as the holotype.

·

Basic reporting

Some references must be modified correctly. Please see my comments on attached PDF in detail.

Experimental design

About identification of the humerus (referred specimen), I am not sure why did you regard it as belonging to the same species with holotype? Please see my comment on attached PDF in detail.

Validity of the findings

About the pedal ungual, it seems to be that of right pes for me. Please see my comment in attached pdf.

Additional comments

I have fully enjoyed your quite important and interesting revision of Sanpasaurus as sauropod, not ornithischian. I am looking forward to seeing this paper published soon.

·

Basic reporting

Article meets standards

Experimental design

Article meets standards

Validity of the findings

See comments below.

Additional comments

General comments:

-I think the material is clearly sauropod, and the discussion does a good job of reinforcing that for the holotype and referred materials.

-Where I am less convinced is with the association of some of IVPP V156B – I think you can clearly associate the dorsal vertebrae, but the rest of it? I have two primary concerns:
1), the nonoverlapping material is just so generic (humerus aside)
2), there’s no paper trail linking this material to a locality (particularly for the darker
material, this is a concern), and given the other potentially relevant material (the
Young, Young and Chow, and Dong referred specimens), I just don’t know.

-That said, I given that the holotype is restricted, I don’t think there’s a problem referring the other material, although having one of the two autapomorphies occur only on the referred material might be problematic down the road, and it might be worth rethinking that decision. The lateral ridges are already pretty weird, and you’re not trying to establish a new taxon, merely distinguish an existing one, so the burden of proof might be considered to be a little lower.

Specific comments:

-Description

1) I’m not sure why you describe the referred dorsals first and the holotype second, but that seems out of order to me.

2) Bearing in mind that I have not seen the material first hand, and you have:

A) I’m not sure I agree about the osteological maturity. The neurocentral junction
seems to be a flat (non-interdigitated), patent suture, which is not indicative of a
late juvenile/subadult osteological age.

B) It’s not clear to me if the TPRL a) meet and/or b) continue ventrally toward the
neural canal. Can you clarify those? I’m just wondering how closely they
approximate the conditions in more derived sauropods.

C) The “thin bony septum” (Lines 252-253): this is confusingly worded, to me. It
wasn’t clear to me whether this was referring to the autapomorphy (which I think it
is not), to the anteroposteriorly oriented ridge dividing the CDF into dorsal and
ventral subfossae, as appears to be the case in Fig. 3E (and maybe 3D, but it’s
hard to tell from the PDF. I can only see it in the original, uncompressed artwork, so
it’s not going to be something most readers will be able to see), or to a midline
(sagittal) septum. It might be worth adding some more descriptive text here
(“oriented along the anteroposterior midline” suggests a subvertical orientation to
me, at least), and maybe labelling the relevant feature on the figure as well. Also,
are you saying that Barapasaurus et al. have a CDF, or that they have a divided
CDF?

i) If it is the lamina dividing the CDF, I would argue that such a condition is very
similar to the posterior dorsal vertebrae of Tazoudasaurus, for example

ii) The CDF is inclusive of the entire area between the diapophysis, the
centrodiapophyseal/parapophyseal laminae, and the centrum; as labeled on the
figure, it looks like you’ve restricted it to the area above the parapophysis.

3) It’s not clear to me if the derived sauropodmorph feature is “tall pedicel”, “tall neural arch”, or “pedicel/neural arch at least 1.5x centrum height” (Lines 234-237).

4) Re: the humerus - It’s true also of the lateral ridge above, but it would be helpful if you matched up the terminology in the text (e.g., median anterodistal projection) with the terminology in the figures (e.g., median tubercle)

-Discussion

1) As I noted above, I think it’s pretty clear this material, whatever it all belongs to, is not ornithischian. The arguments laid out here are pretty convincing.

2) I’m still confused here about the “lateral excavation at the base of the neural arches”, I guess. Is this a CDF? If so, it’s pretty typical, no?

3) I’m not sure how much you can say about biogeography based on a taxon that you refer to Sauropoda incertae sedis. In particular, I don’t think it’s possible to comment on how a taxon might shake up proposed groups when there is explicitly no novel ingroup relationship being suggested. It’s further problematic for me because the only material definitively linked to Sanpasaurus (i.e., the holotype) is not the material providing a lot of the character data used for the biogeography (appendicular material). I think the paper would be better served by removing this section, tbh, but if it’s retained I would be more circumspect about the ultimate implications of the material in question.

---

## Round 0.2 · accepted · Accept

Congratulations. I look forward to seeing your paper published.

·

Basic reporting

No comments

Experimental design

No comments

Validity of the findings

No comments

Additional comments

I'm still unenthused about the biogeography, but I think the changes made are sufficient for publication.